# Mixed Likelihood Variational Gaussian Processes

## Abstract

Gaussian processes (GPs) are powerful models for human-in-the-loop experiments due to their flexibility and well-calibrated uncertainty. However, GPs modeling human responses typically ignore auxiliary information, including a priori domain expertise and non-task performance information like user confidence ratings. We propose mixed likelihood variational GPs to leverage auxiliary information, which combine multiple likelihoods in a single evidence lower bound to model multiple types of data. We demonstrate the benefits of mixing likelihoods in three real-world experiments with human participants. First, we use mixed likelihood training to impose prior knowledge constraints in GP classifiers, which accelerates active learning in a visual perception task where users are asked to identify geometric errors resulting from camera position errors in virtual reality. Second, we show that leveraging Likert scale confidence ratings by mixed likelihood training improves model fitting for haptic perception of surface roughness. Lastly, we show that Likert scale confidence ratings improve human preference learning in robot gait optimization. The modeling performance improvements found using our framework across this diverse set of applications illustrates the benefits of incorporating auxiliary information into active learning and preference learning by using mixed likelihoods to jointly model multiple inputs.

## 1. Introduction

Gaussian process (GPs) are indispensable models for many machine learning and AI applications (Williams & Rasmussen, 2006). As a Bayesian nonparametric model, it is favored for its well-calibrated uncertainty estimates and flexibility. When using a GP for regression with Gaussian

[1]Anonymous Institution, Anonymous City, Anonymous Region, Anonymous Country. Correspondence to: Anonymous Author <anon.email@domain.com>.

Preliminary work. Under review by the International Conference on Machine Learning (ICML). Do not distribute.

errors, i.e., a Gaussian likelihood, the posterior distribution is a multivariate normal whose mean and covariance can be computed analytically via the Kriging equations (Gramacy, 2020). This analytic tractability does not hold for other types of observations, such as classification or preference data, but GP modeling in those settings can be done with variational approximation (e.g., Hensman et al., 2013; 2015) or other approximate inference schemes (Kuss & Rasmussen, 2005).

Human feedback, which is generally non-Gaussian, has recently become an important setting for GP modeling with applications including health screening (Gardner et al., 2015a;b), AR/VR development (Guan et al., 2022; 2023; Kwak et al., 2024), and robot locomotion learning (Tucker et al., 2020). In particular, preference learning has attracted a great deal of attention in recent years for its usefulness in large language model (LLM) training and reinforcement learning with human feedback (RLHF) (Stiennon et al., 2020; Ouyang et al., 2022). GPs are a natural fit for many preference learning problems (Chu & Ghahramani, 2005; Houlsby et al., 2012), including for RLHF (Kupcsik et al., 2018). Due to their well calibrated uncertainty, GPs are especially useful in human-in-the-loop experiments where the human's time is valuable, as GPs can be used with active learning to increase trial efficiency (Owen et al., 2021).

In many non-Gaussian observation settings, multiple data of different types can be observed simultaneously. For example, in preference learning, we can solicit both preferences (binary comparison data) and strengths of preference (e.g., Likert scale survey data). Studies of human perception can measure whether or not a stimulus was perceived (binary classification data) while simultaneously recording response time (continuous but non-Gaussian data). Presumably, combining these different types of data into a single GP would help improve modeling performance. In addition, we may also have domain knowledge about the responses for some special inputs. For instance, in studies of human perception, a stimulus with no intensity cannot be perceived at all. This domain knowledge constraint, as we will show, can be also be considered as an additional observation type that we wish to include in the GP.

Here, we present a framework for joint GP modeling of multiple types of data and expand GP modeling to a rich new set of multi-data-type problems via the following contributions:

- We develop a novel evidence lower bound (ELBO) formulation that includes multiple likelihoods in the same variational approximation.
- We show that our mixed likelihood training can be used to encode domain knowledge in non-Gaussian settings, and thereby accelerate active learning.
- We develop both synthetic and real-world examples of mixed likelihood variational GPs improving model performance by incorporating auxiliary survey data into preference learning and human perception studies, also developing a new Likert likelihood.

## 2. Background

A Gaussian process (GP) $f \sim \mathcal{GP}(0, k)$ defined by a kernel function $k : \mathbb{R}^d \times \mathbb{R}^d \to \mathbb{R}$ is a stochastic process whose function values $\mathbf{f} = f(\mathbf{X})$ on any training data $\mathbf{X} \in \mathbb{R}^{n \times d}$ follow a joint Gaussian distribution $\mathbf{f} \sim \mathcal{N}(\mathbf{0}, \mathbf{K_{f,f}})$, where $\mathbf{K_{f,f}} = k(f(\mathbf{X}), f(\mathbf{X}))$ is the covariance matrix.

A likelihood is a distribution that models the relation between the observed training labels $\mathbf{y}$ and the latent function values $\mathbf{f}$. Thus, different types of data require different likelihoods. For regression, it is common to use a Gaussian likelihood

$$p(\mathbf{y} \mid \mathbf{f}) = \mathcal{N}(\mathbf{y}; \mathbf{f}, \sigma^2 \mathbf{I}),$$

where $\sigma$ is the standard deviation of the label noise. For classification, it is common to use a Bernoulli likelihood

$$p(\mathbf{y} \mid \mathbf{f}) = \mathrm{Bernoulli}(\Phi(\mathbf{f})).$$

On the test data $\mathbf{X}^*$, the GP prediction is the posterior conditioned on the training labels $p(\mathbf{f}^* \mid \mathbf{y})$. For a Gaussian likelihood, the posterior distribution is also Gaussian with a closed-form expressions for its mean and covariance:

$$\mathbb{E}[\mathbf{f}^* \mid \mathbf{y}] = \mathbf{K_{*,f}}(\mathbf{K_{f,f}} + \sigma^2 \mathbf{I})^{-1}\mathbf{y},$$
$$\mathbb{D}[\mathbf{f}^* \mid \mathbf{y}] = \mathbf{K_{*,*}} - \mathbf{K_{*,f}}(\mathbf{K_{f,f}} + \sigma^2 \mathbf{I})^{-1}\mathbf{K_{f,*}}.$$

However, for non-Gaussian likelihoods, the exact posterior is almost always intractable, and thus needs approximation.

Variational GPs (e.g., Titsias, 2009; Hensman et al., 2013; 2015) approximate the exact posterior by inducing point approximation and variational inference (Blei et al., 2017). A set of inducing variables $\mathbf{u}$ is introduced, and the joint distribution factors as follows:

$$p(\mathbf{f}, \mathbf{f}^*, \mathbf{u}) = p(\mathbf{f} \mid \mathbf{u})p(\mathbf{f}^* \mid \mathbf{u})p(\mathbf{u}),$$

where each component admits the form

$$p(\mathbf{u}) = \mathcal{N}(\mathbf{u}; \mathbf{0}, \mathbf{K_{u,u}}),$$
$$p(\mathbf{f} \mid \mathbf{u}) = \mathcal{N}(\mathbf{f}; \mathbf{K_{f,u}}\mathbf{K_{u,u}^{-1}}\mathbf{u}, \mathbf{K_{f,f}} - \mathbf{K_{f,u}}\mathbf{K_{u,u}^{-1}}\mathbf{K_{u,f}}),$$
$$p(\mathbf{f}^* \mid \mathbf{u}) = \mathcal{N}(\mathbf{f}^*; \mathbf{K_{*,u}}\mathbf{K_{u,u}^{-1}}\mathbf{u}, \mathbf{K_{*,*}} - \mathbf{K_{*,u}}\mathbf{K_{u,u}^{-1}}\mathbf{K_{u,*}}).$$

Given the inducing values $\mathbf{u}$, the latent function values on the training data and the test data are conditionally independent. As a result, the prediction on the test data is completely controlled by the inducing variables.

Inference in variational GPs is performed by maximizing the evidence lower bound (ELBO):

$$\underset{q(\mathbf{u})}{\mathrm{maximize}}\, \mathbb{E}_{q(\mathbf{f})} \log p(\mathbf{y} \mid \mathbf{f}) - \mathrm{D_{KL}}(q(\mathbf{u}), p(\mathbf{u})),$$

where the variational distribution $q(\mathbf{u})$ is usually restricted to a Gaussian family and $q(\mathbf{f}) = \int p(\mathbf{f} \mid \mathbf{u})q(\mathbf{u})\, \mathrm{d}\mathbf{u}$ is the marginalized variational distribution over the latent function values. The optimal variational distribution $q(\mathbf{u})$ is then used to construct the approximate posterior

$$p(\mathbf{f}^* \mid \mathbf{y}) \approx \int p(\mathbf{f}^* \mid \mathbf{u})q(\mathbf{u})\, \mathrm{d}\mathbf{u}.$$

## 3. Mixed Likelihood Variational Inference

Suppose we have $T$ different types of data available

$$\left(\mathbf{X}^{(t)}, \mathbf{y}^{(t)}\right), \quad t = 1, 2, \cdots, T,$$

where $\mathbf{X}^{(t)}$'s are training data locations and $\mathbf{y}^{(t)}$'s are labels of different types. For example, $\mathbf{y}^{(1)}$ could be regression labels while $\mathbf{y}^{(2)}$ are classification labels.

We assume that all labels are generated from the same latent function and that the labels $\mathbf{y}^{(t)}$ are conditionally independent given the latent function values $\mathbf{f}^{(t)} = f(\mathbf{X}^{(t)})$, across data types $t$. We then jointly model the different data types by training a single variational GP on all data using a combined ELBO. As before, we use a variational distribution $q(\mathbf{u})$ to approximate the GP posterior. Let $\mathbf{y} = \{\mathbf{y}^{(t)}\}_{t=1}^T$ now represent the complete collection of training labels across data types, and $\mathbf{f} = \{\mathbf{f}^{(t)}\}_{t=1}^T$ their corresponding latent function values. Because of the conditional independence of the various observations, we have that

$$\log p(\mathbf{y} \mid \mathbf{f}) = \sum_{t=1}^T \log p_t\left(\mathbf{y}^{(t)} \mid \mathbf{f}^{(t)}\right).$$

Each type of data uses a different likelihood. For instance, $p_1(\cdot \mid \cdot)$ is a Gaussian likelihood if $\mathbf{y}^{(1)}$ are regression labels, and $p_2(\cdot \mid \cdot)$ is a Bernoulli likelihood if $\mathbf{y}^{(2)}$ are classification labels. The evidence term in the ELBO thus decomposes, and we can write a valid evidence lower-bound for mixed likelihoods as:

$$\sum_{t=1}^T \mathbb{E}_{q\left(\mathbf{f}^{(t)}\right)} \log p_t\left(\mathbf{y}^{(t)} \mid \mathbf{f}^{(t)}\right) - \mathrm{D_{KL}}(q(\mathbf{u}), p(\mathbf{u})). \quad (1)$$

For the special case of $T = 1$, the ELBO in (1) reduces to the usual variational GP ELBO. The interesting behavior

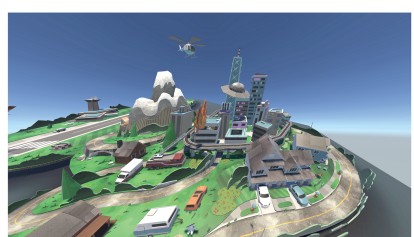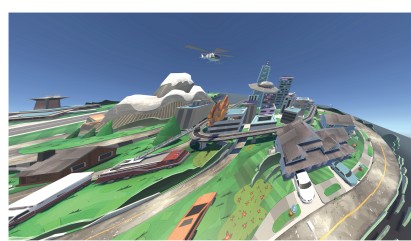

Figure 1: Illustrative depictions of perceived stereoscopic 3D distortions when render cameras are offset from the viewer's eyes. **Left:** Stereoscopic images rendered with cameras at the viewer's eyes have no 3D distortion. **Center:** Small camera offsets result in minimal perceived distortions, and participants cannot reliably identify any errors. **Right:** Large camera offsets result in obvious distortions and are easily recognized. **Bernoulli Level Set Estimation:** If participants are presented the left and middle options in randomized order and asked to select the distorted option, the probability of selecting the correct option will be close to 50%, i.e., at chance. On the other hand, when the left and right options are presented, the distorted option will be selected close to 100% of the time. We aim to identify the space of camera placement configurations such that the distortion detection probability of a participant is below 75%, a threshold that is often considered a just-detectable difference from zero error.

happens when $T > 1$, where multiple types of data are incorporated to learn the same latent function $f$. By virtue of being a valid evidence lower bound, maximizing (1) yields a variational distribution approximating the GP posterior jointly conditioned on multiple types of data.

In the following sections §4 and §5, we demonstrate that this simple idea can solve many problems arising from experimental design and preference learning.

# 4. Encoding Domain Knowledge Constraints in Active Learning

The mixed likelihood training scheme can be used to encode domain knowledge constraints into active learning problems with non-Gaussian data. We demonstrate this in level set estimation with Bernoulli observations, a problem setting with important applications in perception science.

### 4.1. Visual Psychophysics

Understanding human vision and characterizing visual perception is challenging because human self-report is unreliable and individual decision-making criteria are highly variable. Vision scientists use forced-choice experimental paradigms to address these challeneges (Palmer, 1999; Wolfe et al., 2006). Figure 1 describes a psychophysical study design to determine how much render-camera offset is detectable to a person in a stereoscopic (i.e. 3D) display. Rather than asking participants if a particular camera offset looks acceptable, they are given a zero-offset option (the reference) and an option with some offset (the comparison), and asked identify which option has offset. Camera offset is varied over hundreds or thousands of trials with the aim of identifying the set of render-camera offsets that

cannot reliably be differentiated from the zero-offset reference. This is often taken as the offsets for which the probability of correctly selecting the comparison stimulus is below 75% (McKee et al., 1985; Ulrich & Miller, 2004), and this problem can be formulated as Bernoulli level set estimation (Letham et al., 2022).

### 4.2. Bernoulli Level Set Estimation

Given a black-box function $f : \mathbb{R}^d \to \mathbb{R}$, we are concerned with learning the sublevel set $\{\mathbf{x} \in \mathbb{R}^d : f(\mathbf{x}) \leq \gamma\}$ for some constant $\gamma \in \mathbb{R}$. The black-box function $f$ cannot be evaluated directly, but can be "probed" by Bernoulli observations. For any $\mathbf{x} \in \mathbb{R}^d$, we may observe a random variable $y(\mathbf{x}) \in \{0, 1\}$ where

$$y(\mathbf{x}) \sim \text{Bernoulli}(\Phi(f(\mathbf{x}))).$$

We iteratively query the latent function via the Bernoulli observations with the goal of learning the sublevel set. Active learning can be done using a variational GP classification model for $f$, and one of several acquisition functions for proposing new queries (Letham et al., 2022).

The visual psychophysics experiment paradigm described in §4.1 can be cast as Bernoulli level set estimation by taking $\mathbf{x}$ as a visual stimulus and $f(\mathbf{x})$ as the perceptual intensity. We measure, via Bernoulli observations $y(\mathbf{x})$, how well the human participant can differentiate between $\mathbf{x}$ and the reference stimulus $\mathbf{x}_{\text{ref}}$, the perceptual intensity of which is zero. The sublevel set of $f$ is the set of imperceptible stimuli which we wish to identify.

### 4.3. Encoding Prior Knowledge with Soft Constraints

Learning level sets with Bernoulli queries is challenging as Bernoulli observations are inherently noisy, especially for

detection probabilities close to 50%. We generally require many repeated trials in order to accurately estimate the latent response probabilities. Moreover, in the initial stage of active learning, the model is unable to distinguish between stimulus pairs that are obviously different (100% correct) and pairs that are almost identical (50% correct), but guessed correctly. Therefore, we wish to encode *a priori* knowledge about the experimental paradigm.

In the particular case of the visual perception task of §4.1, we know that the detection probability should be exactly 50% when camera offset is 0, and should be close to 100% for maximum offset values. We expect this extra information will improve efficiency of human-in-the-loop experiments with active learning. Moreover, the domain knowledge constraints may add resiliency to prevent active learning from exploring easily-detectable areas away from the target level set in cases when outlier responses occur during the early stage of data collection, e.g., accidental misclicks when the camera offset is large, or several consecutive correct detections by chance even though the camera offset is small.

We encode this type of domain knowledge as constraints on the latent function by directly regressing against the target constraint values. In addition to the Bernoulli observations $(\mathbf{X}^{(1)}, \mathbf{y}^{(1)})$, we produce a set of regression labels $(\mathbf{X}^{(2)}, \mathbf{y}^{(2)})$ provided by domain experts that encode the known latent function values at special locations—for instance, the latent function value should be zero where the detection probability is known to be 50%. We enforce the soft constraints

$$f(\mathbf{x}_i^{(2)}) \approx y_i^{(2)}$$

by mixing Bernoulli with Gaussian likelihoods:

$$p\left(y_i^{(2)} \mid f\left(\mathbf{x}_i^{(2)}\right)\right) = \mathcal{N}\left(y_i^{(2)}; f\left(\mathbf{x}_i^{(2)}\right), \sigma_i^2\right),$$

where $\sigma_i$'s are fixed noise standard deviations that control the softness or hardness of the constraints. Intuitively, we use the Bernoulli likelihood to fit binary responses and the Gaussian likelihood to enforce the constraints.

Figure 2 shows an example of enforcing constraints on an one dimensional objective by mixing Bernoulli and Gaussian likelihoods ($\sigma^2 = 0.001$). Mixed likelihood modeling leads to a significant reduction in posterior uncertainty, as regression labels at the points with known value provide stronger learning signals than the Bernoulli observations.

## 4.4. Synthetic Experiments

We show that enforcing constraints effectively encodes domain knowledge and improves active learning for Bernoulli level set estimation.

We benchmark on three synthetic latent functions. The first is a synthetic two-dimensional psychometric discrimination

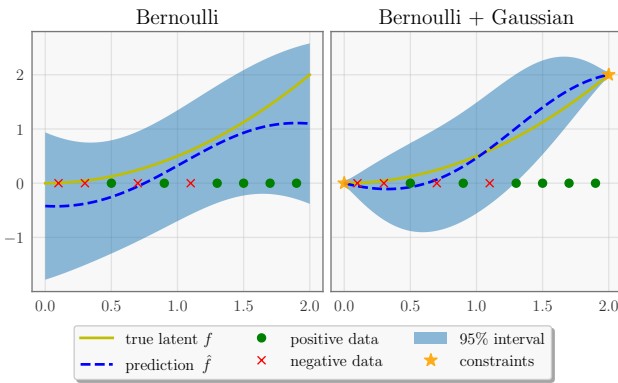

Figure 2: **Left:** A standard variational GP fit to Bernoulli observations. **Right:** A mixed likelihood GP trained on the same data with two constraints $f(0) = 0$ and $f(2) = 2$. The mixed likelihood-trained GP has near-zero uncertainty at the constraint locations. The true latent function is $1/2 \cdot x^2$.

objective from Letham et al. (2022). The others are scaled norm functions $2\|\mathbf{x}\|$ in 2D and 4D respectively. All of these synthetic functions have locations where response probabilities equal exactly 50%, and locations where the response probability is close to 100% probabilities. We use mixed likelihood training to set constraints at a subset of these locations—see §A.1 for more details on the functions and the constraint locations.

We set the Gaussian likelihood noise, which determines strength of the constraint, according to the target value $y_i$ as: $\sigma_i = 0.2 \cdot y_i + 0.1$. Intuitively, this allows a 20% relative violation plus 0.1 absolute violation. For a constraint with $y_i = 0$ (i.e. 50% response probability), this implies an *a priori* 95% credible interval on the response probability of $[0.422, 0.578]$. For $y_i = 2$ (i.e. a 98% response probability), the credible interval for the response probability is $[0.846, 0.999]$. This policy was not extensively tuned, but produces a desirable behavior of maintaining soft constraints across the range of response probabilities. Our preliminary experiments indicate that enforcing constraints with strict tolerances, e.g., $\sigma_i^2 = 10^{-4}$, does not necessarily improve active learning performance as the variational GP tends to be rigid and adapts to new Bernoulli observations slowly. The GP spends most of its prediction capacity fitting the constraints, while spending less weight on Bernoulli observations. This is especially detrimental to look-ahead acquisition functions that depend on the change in posterior conditioning on virtual data.

A natural alternative approach that we use as a baseline is to add Bernoulli pseudo data to the standard single-likelihood GP, to push predictions at those locations towards the target constraint values. Pseudo data are added before active learning, and are added at the same locations used for constraints in the mixed likelihood model. Locations with a response

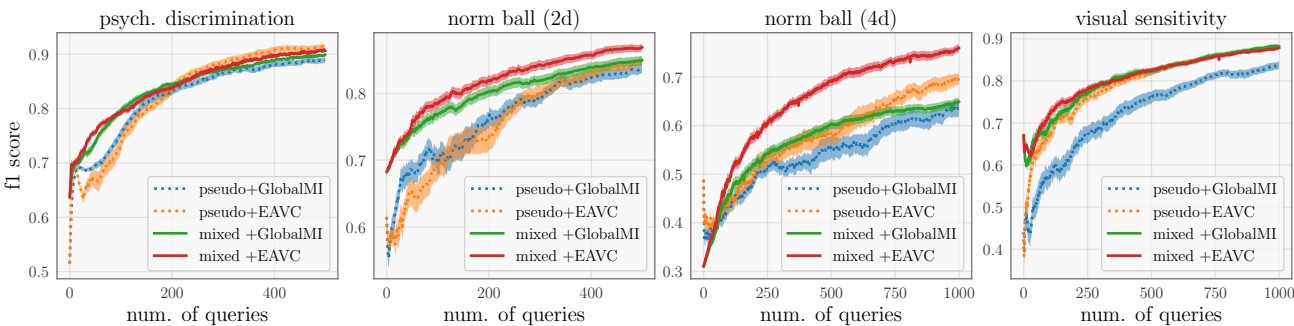

Figure 3: F1 scores (higher is better) of active learning for sublevel set estimation using two different acquisition functions (GlobalMI and EAVC). Domain knowledge for each problem is added either by mixing Bernoulli and Gaussian likelihoods (solid lines) or by adding Bernoulli pseudo data (dotted lines). For both acquisition functions, incorporating domain knowledge with mixed likelihoods led to better F1 scores than the pseudo data approach. Shaded areas show one standard errors over 100 different random seeds.

probability of 50% were given 5 positive and 5 negative Bernoulli data, while those with response probability close to 100% were given a single positive data.

We run active learning for Bernoulli level set estimation with two competitive acquisition functions developed by Letham et al. (2022): global mutual information (GlobalMI) and expected absolute volume change (EAVC). The models were seeded with 10 Sobol-sampled trials before active learning. We evaluate performance by evaluating F1 score of how well the model identifies the sublevel set (with 75% detection probability) at each iteration. F1 score was chosen rather than accuracy because the ground-truth sublevel set only covers a small fraction of the domain, leading to label imbalance typical of real visual psychophysics experiments.

Figure 3 shows active learning performance with four different combinations of models (pseudo-data *vs*. mixed likelihood) and acquisition functions (GlobalMI *vs*. EAVC). For both acquisition functions, imposing domain knowledge via the mixed likelihood framework is better for active learning than the heuristic pseudo data approach.

### 4.5. Mixed-Reality Video Passthrough

We now evaluate the efficacy of the mixed likelihood training for imposing domain knowledge on non-Gaussian observations in a real-world experiment. The real experiment was to measure visual sensitivity to video passthrough camera displacements in a head-mounted display (HMD). Video passthrough is a feature that uses cameras on the outside of an HMD to enable interacting with the world while wearing the device. The passthrough cameras are physically displaced from the user's actual eye position, resulting in inaccurate 3D perception and erroneous motion of the world when the user moves (Biocca & Rolland, 1998).

There are three parameters of interest in this experiment: (a)

differences between camera separation and user interpupillary distance; (b) camera z-axis offsets from the user's eyes due to headset thickness; and (c) passthrough latency, which results in delays between when an image is captured by the cameras and when it is actually seen by a user.

A vision scientist helped us set up a psychophysical experiment to identify the combinations of these three parameters that cannot be reliably differentiated from rendering at the user's actual eye position. We used virtual content and render cameras in order to adaptively explore camera placement relative to viewer eye position, as opposed to real passthrough cameras which would require making changes to physical hardware. They consented to data collection, and collected 900 Bernoulli observations which we analyze in this section. We fit a zero-centered parametric ellipsoid[1] with a sigmoid link function on the collected data:

$$y(\mathbf{x}) \sim \text{Bernoulli}\big(s(\mathbf{x}^\top \mathbf{W}\mathbf{x})\big), \quad \mathbf{x} \in \mathbb{R}^3,$$

where $s(\cdot)$ is a sigmoid function and $\mathbf{W} \in \mathbb{S}^3_{++}$ a symmetric positive definite matrix. This fitted parametric ellipsoid was treated as the ground truth for our model evaluation here. We ran active learning to identify the sublevel set of the 75% detection probability: $\{\mathbf{x} \in \mathbb{R}^3 : \mathbf{x}^\top \mathbf{W}\mathbf{x} \leq s^{-1}(0.75)\}$. Note that this is a slightly misspecified problem—the data generation process is not exactly the same as the model assumption, as the parametric ellipsoid used a sigmoid link function, not the normal CDF link function used by the Bernoulli likelihood.

A total of 21 constraints are imposed by mixed likelihood training: one constraint at the origin $\mathbf{x} = \mathbf{0}$ where the output probability is 50%, and 20 constraints sampled from the domain boundary where the output probability is close to 100%. We again add Bernoulli pseudo data for the standard

---

[1]A quarter ellipsoid to be precise, since headset thickness and latency have to be nonnegative.

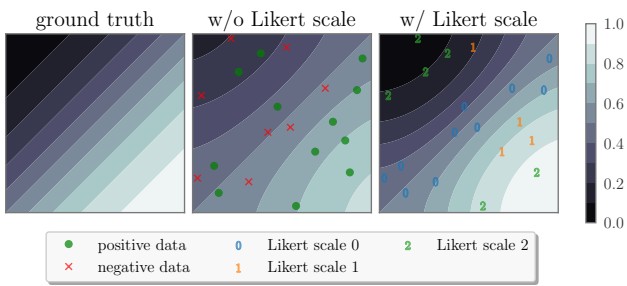

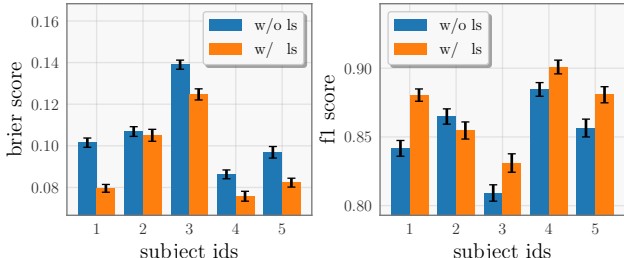

| | | | | |
|---|---|---|---|---|
| ● positive data | **0** Likert scale 0 | **2** Likert scale 2 | | |
| ✕ negative data | **1** Likert scale 1 | | | |

Figure 4: Preference probabilities $\Pr(x_1 \succeq x_2)$ predicted by GPs trained on synthetic data. **Left:** The ground truth probabilities $\Phi(x_1 - x_2)$. **Mid:** A standard variational GP trained on preference observations only, which tends to be under confident at top left and bottom right corners. **Right:** A mixed likelihood GP trained on the same data but with additional synthetic Likert scale ratings on a scale of 0 to 2.

Figure 5: The Brier scores ($\downarrow$) and F1 scores ($\uparrow$) of GPs trained on haptic data collected from five participants. The error bars show one standard error. Mixed likelihood GPs that include Likert scale confidence ratings generally achieve lower Brier scores and higher F1 scores.

variational GP to serve as a baseline. More details of this problem are deferred to §A.1.

In Figure 3 the last panel "visual sensitivity," we plot F1 scores across active learning queries. Mixed likelihood constraints led to higher F1 scores than pseudo data, especially when using the GlobalMI acquisition function, and in early iterations for EAVC.

## 5. Combining User Confidence and Preference

Many vision studies investigate detection thresholds and can be modeled with Bernoulli level set estimation because there is a clearly-defined correct/incorrect user response. However, in many other domains of research related to human perception researchers often evaluate subjective user preferences with psychopysics where there may not be an objectively correct response (Maloney & Yang, 2003; Bartoshuk, 1978; Jones & Tan, 2012). These problems are ideal candidates for preference learning techniques and we next explore how Likert scale survey responses can be used to improve studies in this domain.

### 5.1. Preference Learning and the Likert Likelihood

Given two stimuli $\mathbf{x}_1, \mathbf{x}_2 \in \mathbb{R}^d$, the preference likelihood models the probability that $\mathbf{x}_1$ is preferred to $\mathbf{x}_2$ as

$$\Pr(\mathbf{x}_1 \succeq \mathbf{x}_2) = \Phi(f(\mathbf{x}_1) - f(\mathbf{x}_2)),$$

where $\Phi(\cdot)$ is the normal CDF. The difference of GPs, $f(\mathbf{x}_1) - f(\mathbf{x}_2)$, is itself a GP, hence it is common to directly learn the difference as a GP with a special preference kernel (Houlsby et al., 2011). Then, fitting GPs on preference data is reduced to GP classification.

Suppose that in addition to asking whether $\mathbf{x}_1$ is preferred to $\mathbf{x}_2$ we also ask for a rating of the strength of preference. For example, participants can be asked to rate their confidence on a scale of 1 to 10 after indicating which of the two stimuli is the preferred choice. This differentiates between situations where $\mathbf{x}_1$ is strongly preferred to $\mathbf{x}_2$ or only marginally better. With mixed likelihood training, we can model Likert scale survey responses alongside preference observations to more effectively learn a user's underlying latent function.

We propose a novel likelihood for Likert scale survey responses. Let $y \in \{1, 2, \cdots, l\}$ be the the Likert scale response, with $l \in \mathbb{N}$ the number of options. We call the absolute value of the difference $|f(\mathbf{x}_1) - f(\mathbf{x}_2)|$ the preference strength. Intuitively, high strength preference is correlated with larger Likert scale response. We define cut points,

$$0 = c_1 \leq c_2 \leq, \cdots, \leq c_l < c_{l+1} = \infty,$$

that divide all nonnegative numbers into $l$ intervals,

$$I_i = [c_i, c_{i+1}), \quad i = 1, 2, \cdots, l.$$

Each interval $I_i$ corresponds to a response option. We construct the likelihood so that the probability of observing $y = i$ is highest when the preference strength $|f(\mathbf{x}_1) - f(\mathbf{x}_2)|$ falls into the corresponding interval. We also wish for the likelihood of $y = i$ to be negatively correlated with the distance from the preference strength to the corresponding interval. Hence, we propose the following Likert scale likelihood:

$$\Pr(y = i \mid f(\mathbf{x}_1), f(\mathbf{x}_2)) = \frac{\exp(-\text{dist}_i)}{\sum_{j=1}^{l} \exp(-\text{dist}_j)}. \quad (2)$$

Here the distance is taken as the minimum possible distance to any point in the interval:

$$\text{dist}_i = \min_{a \in I_i} \Big| |f(\mathbf{x}_1) - f(\mathbf{x}_2)| - a \Big|, \quad i = 1, 2, \cdots, l.$$

The cut points $c_i$ in the likelihood are learned automatically by maximizing the ELBO in (1) along with the variational

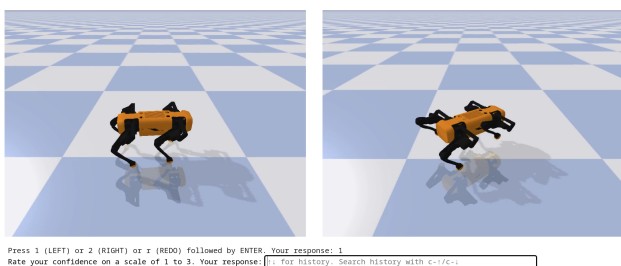

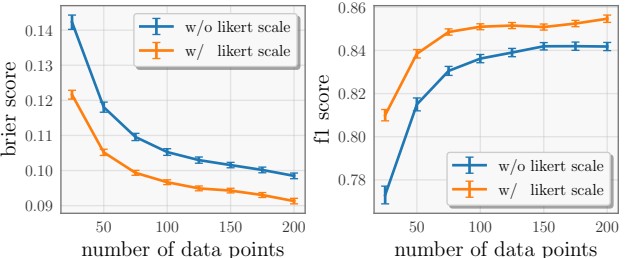

Figure 6: The robot gait data collection GUI. After watching two videos side by side, the human subject reports which robot walks more naturally, and their confidence.

Figure 7: The Brier scores (↓) and F1 scores (↑) of GPs trained on robot gait human evaluations. Error bars show one standard errors. GPs trained with likert scale responses consistently achieve lower Brier scores and higher F1 scores.

parameters on the training data. We add constraints on the cut points to avoid overfitting and a lapse rate parameter to damp the probability (2) with a uniform distribution for enhanced robustness (see §B). Below we apply this Likert scale likelihood on a synthetic and real-world experiments.

One of the challenges of mixing different likelihoods is to ensure they are compatible with each other, in the sense that they can operate on the same latent function. For example, ordinal likelihoods are common for ordinal data, including the Likert scale confidence ratings. However, they cannot be mixed directly with preference observations because they treat the latent function as an "ordinal" strength ranging over entire real numbers. On the other hand, it is the absolute value of the latent difference $f(\mathbf{x}_1) - f(\mathbf{x}_2)$ that represents the preference strength. This necessitates our development of a Likert scale likelihood here.

### 5.2. Synthetic Experiments

In this section, we present experiments on a synthetic latent function. The ground truth function is a univariate identity function $f(x) = x$, and the preference observations are Bernoulli observations with

$$\Pr(x_1 \succeq x_2) = \Phi(x_1 - x_2).$$

The Likert scale responses on a scale of 0 to 2 are generated deterministically based on which interval the preference strength $|f(\mathbf{x}_1) - f(\mathbf{x}_2)|$ falls into: $[0, 0.5], [0.5, 1], [1, \infty)$.

Figure 4 shows GP fits on synthetic data generated from the ground truth latent function. We observe that the GP trained by mixing Bernoulli and Likert scale likelihoods learns the ground truth latent function more accurately, particularly at the corners where the standard GP confidence is too low.

### 5.3. Learning Haptic Preferences

Haptics is the study of how humans perceive the world through the sensation of touch (Kappers & Bergmann Tiest, 2013). We obtained haptic experimental data from Driller (2024, Chapter 5) by contacting the authors. In this study

five participants were presented 50 pairs of 3D printed surfaces with different degrees of microscale surface roughness and material elasticity. For each pair, the user touched both surfaces, reported which felt rougher, and rated the confidence of their judgement on a scale from one to nine.

We split each participant's dataset into 20 training trials and 30 test trials (see §C for additional ablation studies). We train preference GPs with and without confidence ratings on the training set and evaluate them on the test set. Because class imbalance is not a concern with preference data, in addition to the F1 score we also measure the Brier score, which is the mean squared error between the predicted and actual probabilities (Brier, 1950). We repeat the process 100 times with different training/test splits. The average Brier scores and F1 scores with one standard errors are shown in Figure 5. Mixed likelihood GPs trained with the Likert scale likelihood achieved lower Brier scores and higher F1 scores for all subjects except for subject #2.

We observe that subject #2's Likert responses were predominately confident ratings: 48% of their confidence ratings were 9 (the highest rating), 84% of their confidence ratings them were equal to or above 7, and no confidence ratings below 3 were reported. As a result, the Likert scale likelihood struggled to learn the cut points for subject #2. In contrast, other subjects' confidence ratings were spread more evenly (see Figure 9). This suggests that a calibration stage could be important for using survey data in preference learning.

### 5.4. Robot Gait Optimization

Preference learning can also be applied to robotics to align robot behavior with desired outcomes (e.g., Tucker et al., 2020). We consider the robot gait optimization task from Shvartsman et al. (2024), who use human preferences to determine optimal motion control parameters that yield naturalistic robot gait. In their experiment, two videos of quadruped robots are played side by side, and study subjects choose the video with their preferred gait (see Figure 6).

We repeated their study protocol, but additionally collected confidence ratings on a scale of 1 to 3 on each trial. One of authors of this paper participated in this experiment and collected 472 preference responses and confidence ratings. In Figure 7, we report Brier and F1 scores on test sets averaged over 100 different train/test splits. GPs that mixed the Likert scale likelihood with the preference likelihood consistently improved both the Brier score and the F1 score.

## 6. Related Work

From the view of probabilistic graphical models, mixed likelihood training conditions the latent variables (the latent GP function values and inducing values) on different types of observations using variational inference. In fact, variational GPs have been *implicitly* trained with mixed likelihoods in several applications throughout the years. For example, the heteroscedastic Gaussian likelihood, which assigns different noise levels to different data points, is technically a form of mixed likelihood training (Kersting et al., 2007; Lázaro-Gredilla & Titsias, 2011; Binois et al., 2018). Another example is the OR-channel likelihood for modeling multi-tone response in audiometry (Gardner et al., 2015b). Different parameters (number of tones) of the OR-channel likelihood correspond essentially to different likelihoods, and thus it is also an example mixing likelihoods, though using a Laplace approximation and not variational inference.

Importantly, these past examples are all mixtures of likelihoods from the same family. We step further in this paper and introduce a framework for mixing vastly different likelihoods. Recently, Shvartsman et al. (2024) have developed response time GPs that are jointly trained on human choices and response time, but their approach is a single likelihood modeling the joint distribution of both human choices and response time. Their likelihood is based on an approximation of the diffusion decision model designed by domain experts, which is not easily generalizable to other likelihoods or data types. Murray & Kjellström (2018) mixed likelihoods specifically for unsupervised representation learning in GP latent value models. Our work provides a general approach that, as we show, solves many problems arising from experimental designs and preference learning.

Mixed likelihood variational training is closely related to multitask GPs, where the goal is to learn *multiple* correlated latent functions (e.g., Bonilla et al., 2007a;b). Inter-task correlations can be encoded either via Kronecker kernel matrices or, more commonly for variational GPs, using the linear model of coregionalization (LMC) (Alvarez et al., 2012), in which the prediction for each task is a linear combination of multiple variational GPs. Prior work has constructed multi-task models with different likelihoods, each of which is associated with several latent functions, based on both Kronecker and LMC models (Pourmohamad & Lee,

2016; Moreno-Muñoz et al., 2018). Our mixed likelihood approach learns a single, shared latent function from multiple data types. Merging all information into a single GP is necessary for the real-world applications we demonstrate here, such as enforcing domain knowledge constraints.

In §4 we developed domain knowledge constraints as a use case of mixed likelihood modeling. There are other ways to enforce constraints in variational GPs. Recently, Cosier et al. (2024) proposed enforcing constraints with a set of fixed inducing points with fixed inducing values. Compared to this approach, mixed likelihood training has two advantages. First, mixed likelihood training supports soft constraints by tuning the Gaussian likelihood noise, whereas fixing inducing values generally enforces hard constraints. Second, mixed likelihood training is easier to implement, as it is compatible with all off-the-shelf GP variational inference implementations. The only change needed is the training objective. In contrast, fixing inducing values requires a custom implementation of GP variational inference, more specifically a custom whitening strategy.

## 7. Discussion

We have shown that variational GPs can be trained with mixed likelihoods to incorporate multiple types of data in human-in-the-loop experiments. We demonstrated two main applications of mixed likelihood training in this paper: (a) imposing soft constraints on the latent function into GPs by mixing Gaussian likelihoods with Bernoulli likelihoods, and using the constrained variational GPs to accelerate active learning for Bernoulli level set estimation; and (b) leveraging Likert scale confidence ratings by mixing with a Likert scale likelihood to improve preference learning.

A few extensions to our framework are possible. Response time in human-in-the-loop experiments is typically correlated with preference strength, i.e., longer the response time often implies more uncertainty. With an appropriate likelihood for response times, which naturally shares the same latent function with preference observations, mixed likelihood training could significantly simplify the expert-designed likelihood of Shvartsman et al. (2024) based on the diffusion decision model.

Confidence ratings could also be used in active learning with mixed likelihood training, though this comes with challenges. As discussed in §5.3, some participants produce low-quality ratings, which may require calibration (on the fly) before feeding them into the model. Furthermore, asking for confidence ratings increases the cognitive load on participants and may increase the experiment time per trial. It may be best to collect confidence ratings only in the early stage of active learning, when model uncertainty is highest.

## Impact Statement

This paper presents work whose goal is to advance the field of machine learning. There are many potential societal consequences of our work, none of which we feel must be specifically highlighted here.

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

## A. Experimental Details of Bernoulli Level Set Estimation

**Inudcing Points.** All inducing points are fixed, not learned in hyperparameter optimization, because otherwise the GP models overfit easily on the problems we consider. Two types of inducing points are used in the GP models: (a) 100 Sobol samples from the domain; and (b) an inducing point at every constraint location. Both the standard variational GP and the mixed likelihood-trained GP use the same set of inducing points. Crucially, we found the additional inducing points at constraint locations are especially important for mixed likelihood-trained GP. Without these inducing points, the mixed likelihood-trained GP tends to be inflexible.

**Evaluation Metric.** Different from the prior work Letham et al. (2022), which primarily use the Brier score as the evaluation metric, we use the F1 score to evaluate active learning performance for Bernoulli level set estimation.

The Brier score for level set estimation used by Letham et al. (2022) is defined as

$$\frac{1}{n} \sum_{i=1}^{n} (p_i - o_i)^2,$$

where $p_i \in [0, 1]$ is the model's probability prediction on a set Sobol samples in the domain and $o_i \in \{0, 1\}$ is the ground truth indicating whether each $\mathbf{x}_i$ belongs to the sublevel set $\{\mathbf{x} \in \mathbb{R}^d : f(\mathbf{x}) \leq \gamma\}$ for the threshold $\gamma \in \mathbb{R}$. Here, the model's probability prediction for sublevel set is

$$p_i = \Pr(f(\mathbf{x}_i) \leq \gamma) = \Phi\left(\frac{\gamma - \mu_{\mathcal{D}}(\mathbf{x}_i)}{\sigma_{\mathcal{D}}(\mathbf{x}_i)}\right),$$

where $\mu_{\mathcal{D}}$ is the posterior mean and $\sigma_{\mathcal{D}}$ is the posterior standard deviation.

It is often the case, especially in high dimensions, that only a small portion of the domain will have values below the target level set, i.e., the ground truth sublevel set is a tiny fraction of the entire domain. As a result, the vast majority of the ground truths $o_i$ are 0's, which results in label imbalance. For some high dimensional problems, we observe that the Brier score is not reliable due the label imbalance issue, i.e., predicting constant zero might even achieve lower Brier scores than active learning in some cases.

Thus, we opt for the F1 scores evaluated on a set of Sobol samples in the domain for Bernoulli level set estimation. Let $V \subseteq \mathbb{R}^d$ be the ground truth sublevel set and $\widehat{V} \subseteq \mathbb{R}^d$ be the estimate. In the context of level set estimation, the precision and recall have clear geometric interpretations:

$$\text{precision} = \frac{|V \cap \widehat{V}|}{|\widehat{V}|}, \ \text{recall} = \frac{|V \cap \widehat{V}|}{|V|}.$$

We use $10^6$ Sobol samples in the domain to estimate the F1 scores.

**Level Set Estimation Threshold.** Every Bernoulli level set estimation problem in this paper aims for a target sublevel set with a threshold of 75% in the probability space. Equivalently, this is the same as estimating the $\Phi^{-1}(0.75)$ sublevel in the latent function value space

$$\{\mathbf{x} \in \mathbb{R}^d : f(\mathbf{x}) \leq \Phi^{-1}(0.75)\}.$$

The only exception is the parametric ellipsoid in the visual sensitivity task (see §4), where the target sublevel set in the latent function space is

$$\{\mathbf{x} \in \mathbb{R}^d : f(\mathbf{x}) \leq s^{-1}(0.75)\}, \quad s(z) = \frac{1}{1 + \exp(-z)}.$$

This difference is because the parametric ellipsiod uses a sigmoid link function, not the normal CDF.

**Additional Details.** Global look-ahead acquisition functions like GlobalMI and EAVC proposed by Letham et al. (2022) require a set of global reference points. Those reference points are Sobol samples in the domain for estimating global changes in mutual information and sublevel set volumes. We use $10^4$ Sobol samples as reference points. Active learning starts with 10 initial Sobol samples.

### A.1. Objectives

**Psychometric Discrimination.** This function is defined as

$$f(x_1, x_2) = \frac{1 + x_2}{0.05 + 0.4x_1^2(0.2x_1 - 1)^2}$$

on a domain $(x_1, x_2) \in [-1, 1]^2$. It is clear that the Bernoulli probabilities $\Phi(f(\mathbf{x}))$ are exactly 50% on the line

$$\{(x_1, x_2) : -1 \leq x_1 \leq 1, \ x_2 = -1\},$$

and the Bernoulli probabilities are close to 100% on the line

$$\{(x_1, x_2) : -1 \leq x_1 \leq 1, \ x_2 = +1\}.$$

A total of 20 constraints are added: 10 points on the line $x_2 = -1$ and another 10 points on the line $x_2 = +1$. The constraint target values are set to the ground truth latent function values.

**Norm Ball.** This function is defined as

$$f(\mathbf{x}) = 2\|\mathbf{x}\|$$

on a domain $\mathbf{x} \in [-1, 1]^d$. In the main paper, we have used $d = 2$ and $d = 4$. Note that there is a multiplication coefficient 2. The factor 2 makes sure that the function grows fast enough so that the Bernoulli probability $\Phi(f(\mathbf{x}))$ is close to $100\%$ on the domain boundary. We impose a constraint at the origin $\mathbf{x} = \mathbf{0}$ and additionally sample 5 Sobol samples as constraint locations from every hypercube face. The constraint target values are set to the ground truth latent function values.

**Parametric Ellipsoid.** This is a 3D function defined as

$$f(\mathbf{x}) = \mathbf{x}^\top \mathbf{W} \mathbf{x},$$

where

$$\mathbf{W} = \begin{pmatrix} +0.00345447 & -0.00344695 & -0.00144475 \\ -0.00344695 & +0.00556409 & +0.00252343 \\ -0.00144475 & +0.00252343 & +0.00466492 \end{pmatrix}$$

and the domain is $[-30, 50] \times [0, 60] \times [0, 75]$ with each axis being IPD offsets, camera z-axis errors, and passthrough latency. Note that $\mathbf{W}$ is symmetric and positive definite. The weight matrix $\mathbf{W}$ is estimated by maximum likelihood on the collected human data with convex optimization. Note that the link function for this objective is a sigmoid function $s(\cdot) = 1/(1 + \exp(-\cdot))$, not a normal CDF. We impose a constraint at the origin $\mathbf{x} = \mathbf{0}$ and sample 5 Sobol samples as constraint locations from each of the following faces

$$F_0 = \{(x_1, x_2, x_3) : x_1 = -30, \ 0 \leq x_2 \leq 60, \ 0 \leq x_3 \leq 75\},$$
$$F_1 = \{(x_1, x_2, x_3) : x_1 = 50, \ 0 \leq x_2 \leq 60, \ 0 \leq x_3 \leq 75\},$$
$$F_2 = \{(x_1, x_2, x_3) : -30 \leq x_1 \leq 50, \ x_2 = 60, \ 0 \leq x_3 \leq 75\},$$
$$F_3 = \{(x_1, x_2, x_3) : -30 \leq x_1 \leq 50, \ 0 \leq x_2 \leq 60, \ x_3 = 75\},$$

which are four faces that do not contain the origin. The constraint target value at a location $\mathbf{x}$ is set to $\Phi^{-1}(\min\{s(\mathbf{x}), 0.999\})$. Namely, we first evaluate the ground truth probability $s(\mathbf{x})$, then truncate it by $99.9\%$, and then covert it into the corresponding latent function value as if the link function was the normal CDF $\Phi(\cdot)$. The ground truth latent function values cannot be directly used in mixed likelihood training because of the link functions mismatch with each other. Thus, the conversion is necessary. The truncation is also necessary, because otherwise $\Phi^{-1}(s(\mathbf{x}))$ might be infinity due to floating point overflow. Note that this is an example that the "believed" latent function value, not the ground truth latent function value, is used in constraints.

## B. Additional Details of the Likert Scale Likelihood

A constraint $c_{i+1} - c_i \leq 2$ is enforced for each pair of adjacent cut points to avoid overfitting. Note that $\Phi(2) \approx 0.98$. This constraint enforces that the preference probability within the same Likert scale response is no greater than 98%, a natural

assumption on the Likert scale response. With limited number of data, the cut points and the latent function may not be learned accurately. Thus, it is important to add constraints on the cut points to avoid overfitting.

In addition, we introduce a lapse rate parameter to damp the Likert scale likelihood. Let $p_1, p_2, \cdots, p_l$ be the probabilities produced by the Likert scale likelihood (2). Then, we damp the probabilities with a mixutre of uniform distribution

$$p_i^{\text{damp}} = (1 - \lambda)p_i + \lambda \cdot \frac{1}{l},$$

where $\lambda \geq 0$ is the lapse rate parameter. The damped probability is used to train variational GPs in the experiments, and we use a lapse rate of $\lambda = 0.1$ throughout. Intuitively, the damped Likert scale likelihood is more robust because it prevents extremely small probabilities, particularly when the number of data is limited.

## C. Learning Haptic Preferences

The raw Likert scale confidence ratings on a scale of 1 to 9 are mapped to a scale of 0 to 2:

$$1, 2, 3 \mapsto 0, \quad 4, 5, 6 \mapsto 1, \quad 7, 8, 9 \mapsto 2,$$

which is primarily due to the easy of programming.

The Brier score shown in Figure 5 is defined as

$$\frac{1}{n} \sum_{i=1}^{n} (p_i - o_i)^2, \tag{3}$$

where $o_i \in \{0, 1\}$ indicating which stimulus ($\mathbf{x}_{i1}$ or $\mathbf{x}_{i2}$) is preferred and $p_i$ is the GP probability prediction

$$\Phi\left(\frac{\mu_{\mathcal{D}}(\mathbf{x}_{i1}, \mathbf{x}_{i2})}{\sqrt{1 + \sigma_{\mathcal{D}}^2(\mathbf{x}_{i1}, \mathbf{x}_{i2})}}\right),$$

where $\mu_{\mathcal{D}}(\mathbf{x}_{i1}, \mathbf{x}_{i2})$ is the posterior mean of the latent difference $f(\mathbf{x}_{i1}) - f(\mathbf{x}_{i2})$ conditioned on the training data $\mathcal{D}$, and $\sigma_{\mathcal{D}}^2(\mathbf{x}_{i1}, \mathbf{x}_{i2})$ is the posterior variance of the latent difference $f(\mathbf{x}_{i1}) - f(\mathbf{x}_{i2})$ conditioned on the training data.

We use 100 Sobol samples as inducing points for both the standard variational GPs and mixed likelihood-trained GPs. In Figure 8, we present the Brier scores and F1 scores of GPs trained with varying number of data points. Since we only have 50 data points per human subject, we only experiment with training sizes of 10, 20, and 30. Likert scale confidence ratings again improve both Brier scores and F1 scores except for subject #2. As discussed in the main paper, it is most likely because subject #2 Likert scale ratings are predominantly confident. In Figure 9, we plot all subjects' confidence rating histograms. There is a clear difference between the histogram of subject #2 and the remaining subjects: subject #2's ratings tend to be more confident then remain subjects.

## D. Robot Gait Optimization

The Brier score presented in Figure 7 is computed similarly as discussed in §C. We use 100 Sobol samples as inducing points for both the standard variational GPs and mixed likelihood-trained GPs. In Figure 10, we plot the distribution of confidence ratings in the data collected from the robot gait optimization task: 218 of them are 1; 176 of them are 2; and 78 of them are 3. The confidence ratings are generally well-balanced.

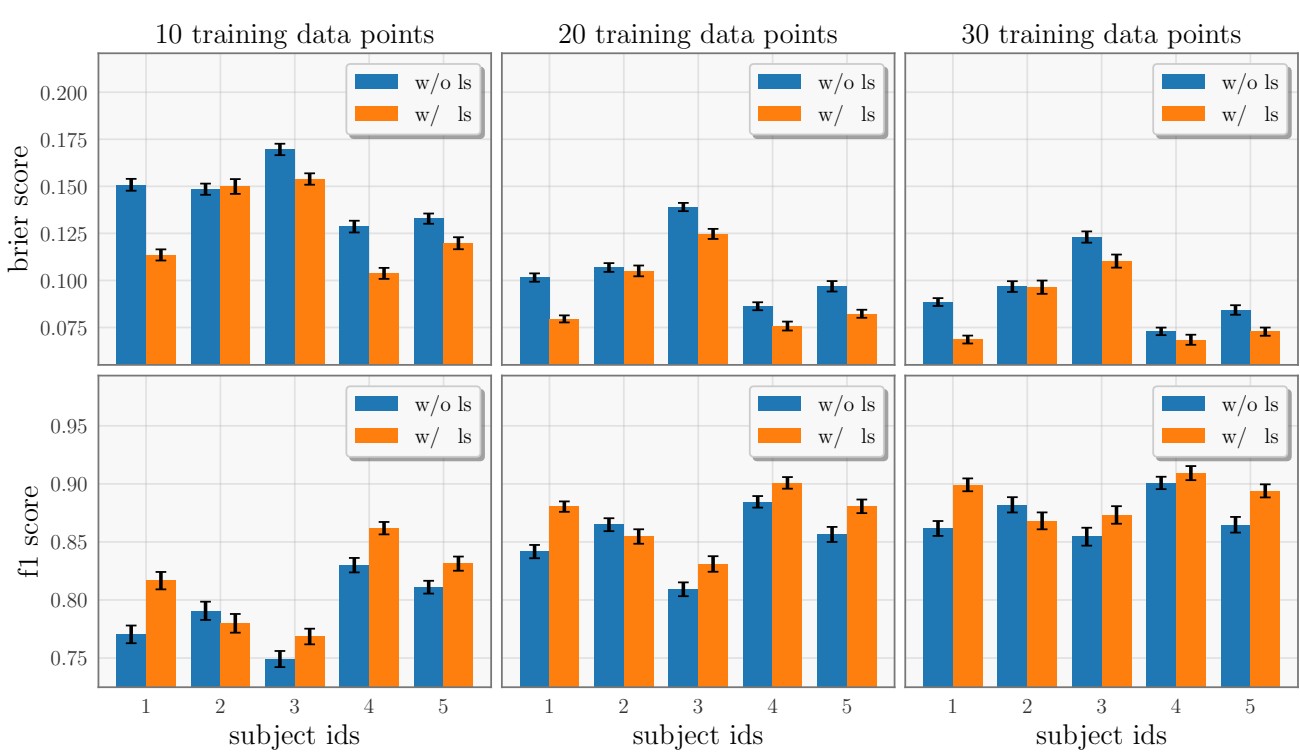

Figure 8: Brier scores and F1 scores of GPs with varying number of training data points on the haptic perception dataset.

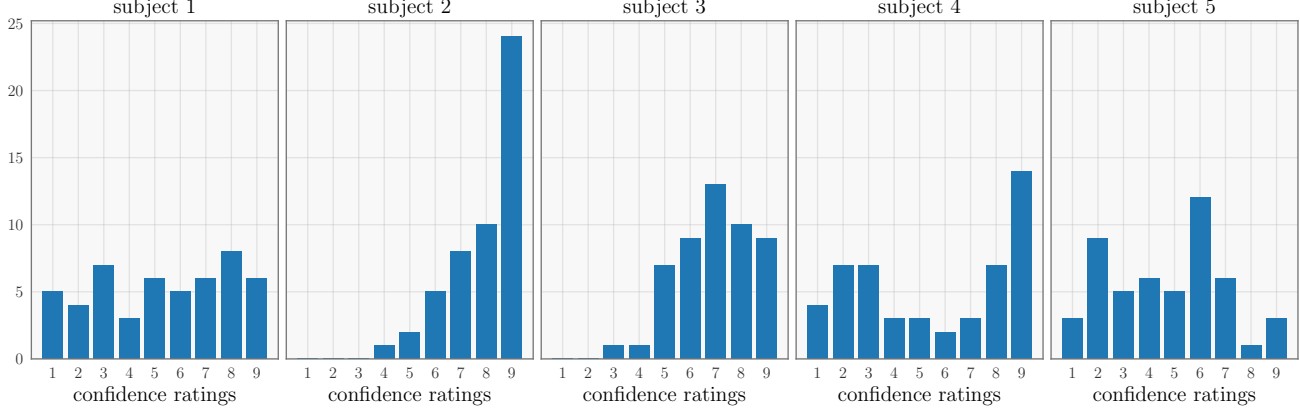

Figure 9: Human subjects' confidence rating histograms in the haptic dataset.

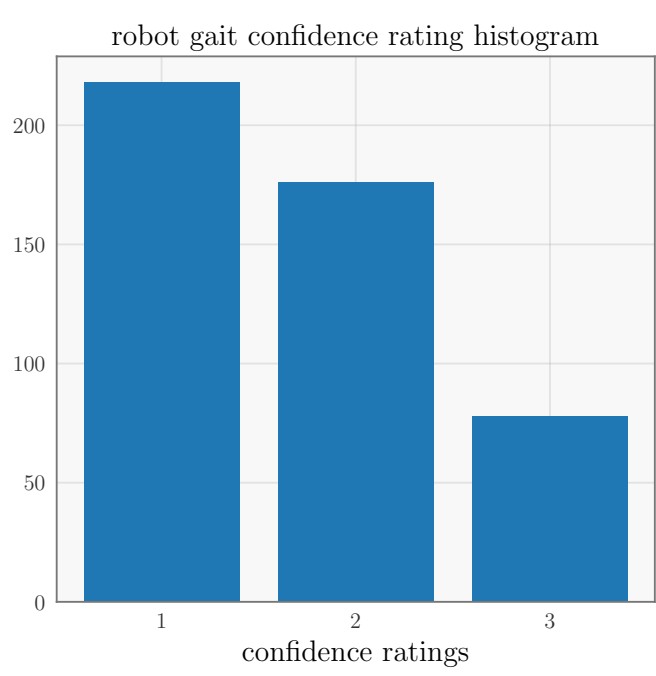

Figure 10: The confidence rating histogram of the data collected from the robot gait optimization task.

