# OpenReview forum: "Mixed Likelihood Variational Gaussian Processes"
_ICML.cc/2025/Conference — Submitted to ICML 2025_

### Official Review · Reviewer_aHaB · 2025-03-08

**Overall Recommendation:** 3

**Summary:**

The paper proposes a method of training a variational gaussian process model with more than one "type" of observations by allowing it to utilise more than one type of likelihood. The authors explain how this method can be used in many real world scenarios, either by enforcing soft-constraints (encoded as additional observations) or by combining different types of information sources.

**Claims And Evidence:**

I believe all claims are properly supported.

**Essential References Not Discussed:**

I believe relevant references are properly discussed.

**Experimental Designs Or Analyses:**

For each experiment authors report the number of seeds and report standard errors, allowing to assess the statistical significance of the results. My (very very minor) suggestion would be to include the number of seeds and type of confidence interval reported (e.g. one standard error) in the descriptions of all Figure rather than only in text, as it makes it easier to find them.

**Methods And Evaluation Criteria:**

I believe authors selected a very interesting and diverse set of real-world experiments, ranging from visual psychophysics, through learning haptic preference, to optimising robot gait. These experiments fit well within the general "story" of the paper and provide a justification for the need of developing the proposed method. My only criticism (and the reason why I only give a score of 3) is the lack of baselines. In all of the experiments authors basically only have two types of methods: the standard version without including auxiliary information and soft constraints and the version with the proposed mixed likelihood that is able to include them. However, authors also admit in Section 6 that there is plenty of similar (but not exactly the same) methods that are either able to include information from different sources (like multi-output GPs) or include constrains (like the work of Cosier et al. 2024). In said section, authors explain the advantage their method has over these baselines, and their explanation is reasonable, however, the paper could be made much stronger by **empirically** showing these advantages by adding them to the comparison in the experiments.

**Other Comments Or Suggestions:**

- I do not like the following sentence in the abstract: "However, GPs modelling human responses typically ignore auxiliary information, including a priori domain expertise ...". One could argue the choice of GP prior (kernel and mean) is precisely supposed to capture a priori domain expertise. I believe abstract could be improved by first mentioning the model and then describing the gap it addresses in existing research, e.g. "Having different likelihood functions gives much more flexibility in encoding soft constraints and auxiliary information than the standard GP model" or something along those lines
- I believe there are much earlier references for the Kriging equations than Gramacy, 2020. For example the famous Gaussian Process book by C.Rasmussen and C.Williams 2006.
- In line 73, I believe the equation of kernel matrix is wrong. The kernel function should operate on the inputs directly that is $ K_{ff} = k(\mathbf{X}, \mathbf{X}) $, rather than on function values.
- The ordering of subsections in section 4 is a bit weird. Section 4.1 introduces the concrete problem, then Sections 4.2 and 4.3 talk about more general concepts and then Sections 4.4 and 4.5 focus again on the concrete problems. The logical flow here is a bit unclear. I think ordering the sections as 4.2, 4.3, 4.1, 4.4, 4.5 would make more sense and would be easier to follow (start with abstract concept -> go to concrete examples)
- On page 3, section 4.2, I believe in the equation, $\Phi(\cdot)$ is not defined. It is defined later as a Gaussian CDF, but it should be defined here, which is where it is used first


**After Rebuttal**
I am generally happy with the paper being accepted. The authors provided some additional experiments, which showcased their method is more effective than multi-output GP in one experiment. Since we still don't have those results for the remaining experiments, I will be keeping my score. Please see my reply rebuttal comment for more information.

**Other Strengths And Weaknesses:**

I particularly like the large number and diversity of relevant real-world experiments. It looks like authors put a lot of effort into gathering the data and it would be great for the overall research community if these datasets could be shared upon acceptance of the paper (preferably with the code allowing to reproduce the experiments). I also really like the detailed analysis of participants preferences in Section 5.3, explaining why the proposed method underperforms for subject 2. In-depth empirical analyses beyond merely quoting the final metric are incredibly valuable.

**Questions For Authors:**

- Why is the focus specifically on human feedback data? I believe the proposed method can employed in other scenarios as well, where auxiliary information or prior knowledge is available. This is by no means a criticism, merely a question.
- Are there any plans to release the code?
- As far as I understand, authors compiled an entirely novel dataset for the purpose of section 4.5. Are there any plans to release that dataset upon acceptance?

**Relation To Broader Scientific Literature:**

As admitted by the authors, many similar methods have been proposed through the years, but none of them is exactly the same and for the suggested applications, the method proposed by the authors seems most suitable. Multi-output GPs are one such example as they are capable of utilising informations from different sources. However, multi-output GPs assume one latent functions per each observation type, whereas the proposed method allows for all observation types to be associated with the same function (instead of observing multiple related functions, we observe one function in "multiple ways"). Also, the proposed method seems to be able to have different number of observations per likelihood type, which would correspond to multi-output GP with missing observations and I believe it is not entirely straightforward to deal with missing observations in multi-output GPs.

Authors also admit that they are not the first to propose mixed likelihood for GPs, but they are the first to do it explicitly with variational inference. Their approach enjoys simplicity and looks like it can be easily integrated with many off-shelf variational GP techniques, making it probably the strongest point of the proposed method.

**Theoretical Claims:**

N/A

---

> ### Author Rebuttal · Authors · 2025-03-31
>
> We thank the reviewer for their constructive comments. Below we provide our response.
>
> > In said section, authors explain the advantage their method has over these baselines, and their explanation is reasonable, however, the paper could be made much stronger by empirically showing these advantages by adding them to the comparison in the experiments.
>
> We have compared mixed likelihood training with heterogeneous multi-output GPs on the robot gait optimization data. The heterogeneous multi-output GPs use two latent GPs, and are trained using two likelihoods: a Bernoulli likelihood to fit the preference observations and a Likert scale likelihood to fit the confidence ratings. The LMC coefficients in the heterogeneous multi-output GPs are learned by maximizing the ELBO on the training data.
>
> We note that heterogeneous GPs have lower Brier scores and higher F1 scores compared to standard variational GPs, especially when more training data is available. However, mixed likelihood trained GPs with a *single* shared latent consistently outperform heterogeneous GPs (with two latents). Moreover, learning two latent GPs and LMC coefficients tends to overfit, especially when the number of training data points is limited; see the F1 scores when \\(n = 25\\) in the right panel.
>
> https://imgur.com/a/4kubKSe
>
> > I do not like the following sentence in the abstract: "However, GPs modelling human responses typically ignore auxiliary information, including a priori domain expertise ...". One could argue the choice of GP prior (kernel and mean) is precisely supposed to capture a priori domain expertise. I believe abstract could be improved by first mentioning the model and then describing the gap it addresses in existing research, e.g. "Having different likelihood functions gives much more flexibility in encoding soft constraints and auxiliary information than the standard GP model" or something along those lines
>
> We acknowledge that some wording in parts of the abstract is inaccurate, and we will modify the abstract accordingly.
>
> > I believe there are much earlier references for the Kriging equations than Gramacy, 2020. For example the famous Gaussian Process book by C.Rasmussen and C.Williams 2006.
>
> Gramacy (2020) is a book that provides an overview of GPs and their applications. We will add a citation to Williams and Rasmussen (2006) here. (We did cite the book by Williams and Rasmussen (2006), but at a different location.)
>
> > In line 73, I believe the equation of kernel matrix is wrong.
>
> Yes, this is a typo. The kernel should be evaluated on the data points, not the function values.
>
> > The ordering of subsections in section 4 is a bit weird. Section 4.1 introduces the concrete problem, then Sections 4.2 and 4.3 talk about more general concepts and then Sections 4.4 and 4.5 focus again on the concrete problems. The logical flow here is a bit unclear. I think ordering the sections as 4.2, 4.3, 4.1, 4.4, 4.5 would make more sense and would be easier to follow (start with abstract concept -> go to concrete examples)
>
> We put Section 4.1 as the first subsection because it motivates the problem of Bernoulli level set estimation, which admittedly many readers may be unfamiliar with. Diving directly into the abstract concepts in Section 4.2 may leave readers unmotivated. Though, we do acknowledge that the ordering could be very well non-optimal. This section was a bit hard to write as we aim to compress a lot of information into it. Nevertheless, we will reassess and polish this section.
>
> > On page 3, section 4.2, I believe in the equation, \\(\Phi(\cdot)\\) is not defined. It is defined later as a Gaussian CDF, but it should be defined here, which is where it is used first.
>
> Thank you for pointing out this. We will add the definition at its first appearance.
>
> > Why is the focus specifically on human feedback data? I believe the proposed method can be employed in other scenarios as well, where auxiliary information or prior knowledge is available. This is by no means a criticism, merely a question.
>
> Yes, we are also interested in finding other applications of the methods. We focus on human feedback data primarily because we have access to these types of data.
>
> > Are there any plans to release the code?
>
> Yes, we plan to release the code upon acceptance.
>
> > As far as I understand, authors compiled an entirely novel dataset for the purpose of section 4.5. Are there any plans to release that dataset upon acceptance?
>
> We plan to release all data used in this paper. The only exception is the haptic preference data in Section 5.3, because this is not collected by us and is out of our hands.

---

> > ### Comment · Reviewer_aHaB · 2025-04-02
> >
> > Thank you for your response. My main concern was related to baselines and the authors have partially addressed it by providing multi-output GP results on one of the experiments. If authors are able to provide results on multi-output GP results on more experiments during the discussion period, I would be happy to increase my score. Since the baseline results so far are only partial, **I am generally happy with the paper being accepted, but I will maintain my score of 3.**
> >
> >
> > In general, if the paper is accepted, I would expect authors to:
> > - include multi-output GP baselines for all experiments, where it is possible to
> > - adjust the wording and polish writing in the parts highlighted in my review
> > - open-source the code and the new dataset

---

### Official Review · Reviewer_jpud · 2025-03-13

**Overall Recommendation:** 4

**Summary:**

This paper introduces mixed likelihood variational Gaussian Processes (GPs) to incorporate auxiliary information by combining multiple likelihoods within a single evidence lower bound. The authors demonstrate the method’s effectiveness across three human-centered experiments: (1) accelerating active learning in a visual perception task by integrating prior knowledge in GP classifiers, (2) improving haptic perception modeling of surface roughness using Likert scale confidence ratings, and (3) enhancing preference learning in robot gait optimization through confidence-rated feedback. Results show consistent modeling improvements, highlighting the value of leveraging auxiliary data through mixed likelihoods in active and preference learning.


## update after rebuttal
I confirm my score and thank the authors for addressing my comments as well as those of the other reviewers.

**Claims And Evidence:**

The claims are well supported by clear evidence, thoughtful analysis, and relevant references to prior work.

**Essential References Not Discussed:**

N/A

**Experimental Designs Or Analyses:**

The experiments are well designed and provide meaningful insights. The results are analyzed clearly and convincingly, and the ablations are focused and effectively highlight the contributions of each component.

**Methods And Evaluation Criteria:**

The proposed methods and evaluation criteria are well aligned with the problem and application domains. The use of mixed likelihoods is well motivated for incorporating auxiliary information, and the chosen tasks—visual perception, haptic perception, and robot gaiting—are interesting and diverse testbeds that effectively demonstrate the benefits of the approach.

**Other Comments Or Suggestions:**

None

**Other Strengths And Weaknesses:**

The paper is well-written and presents an original approach. The claims, theoretical insights, and experimental results are clearly presented, and the findings offer valuable implications for advancing variational GPs, preference encoding, and robotics applications.

**Questions For Authors:**

Could other types of scores/scales play similar role as the Likert scale? If so, did you consider any other in particular? or is there a particular reason behind the choice of the Likert scale? If other scales are not applicable, can you clarify why?

Would your method scale with high dimensional input (eg raw sensor data from a real robot)? If not, which adjustments would be needed?

**Relation To Broader Scientific Literature:**

The key contributions of this paper are relevant to the community, offering valuable insights that could advance work in variational GPs, preference encoding, and potentially embodied behaviors, such as robotic motion.

**Theoretical Claims:**

I reviewed the theoretical claims and equations, although I did not verify all mathematical derivations in detail. While it is possible that I may have overlooked some aspects, the theoretical claims appear to be correct to the best of my understanding.

---

> ### Author Rebuttal · Authors · 2025-03-31
>
> We thank the reviewer for their constructive comments. Below we provide our response.
>
> > Could other types of scores/scales play similar role as the Likert scale? If so, did you consider any other in particular? or is there a particular reason behind the choice of the Likert scale? If other scales are not applicable, can you clarify why?
>
> Yes, there are other types of scales available. For example, we could use a slider scale, where the confidence rating is collected by asking subjects to drag a slider along a continuous scale from 1 to 10. The slider scale rating is continuous but non-Gaussian, as the output range is between 1 and 10. Hence, it requires designing a likelihood for the slider scale. In this paper, we choose the Likert scale due to its simplicity.
>
> > Would your method scale with high dimensional input (eg raw sensor data from a real robot)? If not, which adjustments would be needed?
>
> Whether the model applies to raw sensor data in robotics depends on whether GPs are suitable for these types of data, which is orthogonal to our mixed likelihood training methodology. This could be an interesting direction to explore in the future.

---

> > ### Comment · Reviewer_jpud · 2025-04-02
> >
> > Thank you for addressing my comments. I think the explanations provided would be nice to read in the paper as well.

---

### Official Review · Reviewer_f9ai · 2025-03-14

**Overall Recommendation:** 3

**Summary:**

The paper develops a method for variational Gaussian processes (GPs) using mixed likelihoods, i.e., when for the same input data and latent function there  exist multiple and different kind of output observations. The authors train their model using an evidence lower bound by utilizing also inducing variables to deal with big data.  The application of variational inference is quite straightforward. Then the paper considers applications that involve data in human-in-the-loop experiments. Particularly, the first application involves imposing soft constraints by mixing Gaussian likelihoods and Bernoulli likelihoods, in order to speed up active learning for Bernoulli level set estimation. In a second application the authors combine preference/ranking binary data with scale confidence ratings. For the scale confidence ratings a Likert likelihood is used instead of an ordinal regression likelihood.  Several experimental results show that the method is useful in practice.

**Claims And Evidence:**

The experimental results provide clear evidence that the whole method can be useful for mixed likelihood supervised learning problems.

**Essential References Not Discussed:**

No.

**Experimental Designs Or Analyses:**

The experimental analysis is valid.

**Methods And Evaluation Criteria:**

The evaluation criteria and the benchmarks are very appropriate to experimentally demonstrate the proposed method.

**Other Comments Or Suggestions:**

The paper is very well written. I didn't find any typos.

**Other Strengths And Weaknesses:**

The main strength is the experimental study and the applications. Almost 5 or 6 pages of the paper are devoted to the applications.
Methodologically, regarding the variational method the paper is incremental.

**Questions For Authors:**

One question I have is about the motivation behind the  Likert scale likelihood. Someone would expect the use of ordinal
regression likelihood for the ratings. However, the authors  claim that ordinal likelihoods "cannot
be mixed directly with preference observations because they treat the latent function as an “ordinal” strength ranging over
entire real numbers". Can you clarity further this point?

If we cannot use ordinal likelihoods, then this could mean that in general mixing different likelihoods is not so easy
because modeling different outputs may require modeling different "scales" of the shared GP function. However,
i can imagine that there must be some principled ways to resolve this, such as by introducing some learnable scaling parameter
per likelihood.

**Relation To Broader Scientific Literature:**

This paper essentially builds on the previous literature on multiple output GPs. The related work in the paper covers this connection.

**Theoretical Claims:**

Yes, all derivations in the paper are correct.

---

> ### Author Rebuttal · Authors · 2025-03-31
>
> We thank the reviewer for their constructive comments. Below we provide our response.
>
> > One question I have is about the motivation behind the Likert scale likelihood. Someone would expect the use of ordinal regression likelihood for the ratings. However, the authors claim that ordinal likelihoods "cannot be mixed directly with preference observations because they treat the latent function as an “ordinal” strength ranging over entire real numbers". Can you clarity further this point?
>
> The ordinal likelihood assumes the probability of observing a Likert scale rating \\(y = i\\) is
> \\[
> \Pr(y = i \mid f) = \Phi(c_i - f) - \Phi(c_{i-1} - f) \\; \text{for} \\; i \geq 2, \quad \text{and} \quad \Pr(y = 1 \mid f) = \Phi(c_1 - f) \\; \text{for} \\; i = 1,
> \\]
> where \\(c_i\\) are cut points.
> Note that the latent \\(f\\) has to have a range \\(\mathbb{R}\\).
> In particular, \\(f\\) needs to be able to go to \\(-\infty\\).
> Otherwise, \\(\Pr(y = 1 \mid f) \\) is always lower bounded by a constant, in which case the likelihood is unable to output all categorical distributions.
>
> In preference learning, however, we want to predict the Likert scale ratings based on the preference strength, i.e., the absolute value of the latent function difference \\(\lvert f(x_1) - f(x_2) \rvert\\). The intuition is that larger preference strength correlates with higher confidence ratings. The preference strength is a non-negative number and is not compatible with the ordinal likelihood.
>
> > If we cannot use ordinal likelihoods, then this could mean that in general mixing different likelihoods is not so easy because modeling different outputs may require modeling different "scales" of the shared GP function. However, i can imagine that there must be some principled ways to resolve this, such as by introducing some learnable scaling parameter per likelihood.
>
> Yes, we need to be careful what likelihoods can be mixed together.
> At the end of the day, we need to ensure all likelihoods are able to share the same latent function.
> We did try to use the ordinal likelihood by applying the following transformation on the latent
> \\[
> h(\mathbf{x}_1, \mathbf{x}_2) = \log\big(\lvert f(\mathbf{x}_1) - f(\mathbf{x}_2)\rvert + \epsilon\big),
> \\]
> where \\(\epsilon > 0\\) is a small positive constant.
> Now \\(h(\mathbf{x}_1, \mathbf{x}_2)\\) has a range \\(\mathbb{R}\\), and then we put an ordinal likelihood on \\(h\\).
> However, this trick does not improve the performance.
> We believe it fails due to model misspecification.
> Even though the log transformation fixes the obvious range issue, it imposes a strong assumption that the ordinal strength (how likely the subject chooses a high confidence rating) grows **logarithmically** with respect to the preference strength.
> It is possible that there might be other non-linear transformations that could make the ordinal likelihood work, e.g., adding additional tunable parameters to the above log transformation.
> But the transformation cannot be a simple (linear) scaling.

---

### Official Review · Reviewer_pCFS · 2025-03-20

**Overall Recommendation:** 3

**Summary:**

This paper proposes Mixed Likelihood Variational Gaussian Processes (GPs) as a method to integrate auxiliary information (e.g., domain expertise, confidence ratings) into GP models for human-in-the-loop experiments. Traditional GP models often assume a single likelihood and ignore non-task-specific information. The proposed method addresses this by incorporating multiple likelihoods within a single evidence lower bound (ELBO) formulation, allowing the GP model to jointly model multiple types of human feedback.

The paper provides three real-world applications:
1. Visual Perception Task – Incorporating domain knowledge constraints improves active learning efficiency in identifying camera position errors in virtual reality.
2.  Haptic Perception Task – Using Likert-scale confidence ratings enhances model fitting for surface roughness perception.
3. Robot Gait Optimization – Integrating confidence ratings into human preference learning improves model performance in optimizing robot gait.

**Claims And Evidence:**

Most of the paper’s claims are supported by empirical evidence from real-world experiments. The key claims and their support are as follows:
1. GPs benefit from mixed likelihood modeling – The experiments show improved performance in multiple tasks when auxiliary information is incorporated.
2. Domain knowledge can be effectively integrated into GPs – The visual perception task demonstrates how prior knowledge constraints accelerate active learning.
3. Confidence ratings improve human-in-the-loop learning – Both haptic perception and robot gait optimization tasks show better fitting when Likert-scale ratings are included.

**Essential References Not Discussed:**

No obvious missing.

**Experimental Designs Or Analyses:**

The experiments cover three diverse real-world tasks, making the results more generalizable.

**Methods And Evaluation Criteria:**

The evaluation metrics (e.g., improved model fitting and active learning efficiency) are relevant for human-in-the-loop learning.

**Other Comments Or Suggestions:**

No.

**Other Strengths And Weaknesses:**

Strengths:
** The paper presents novel contributions to GP modeling by effectively integrating auxiliary information (e.g., confidence ratings, domain knowledge).
** The experiments are diverse and showcase the generality of the approach across different tasks. The use of real-world data adds credibility to the results.
** The paper is well-written and clear, with a logical progression from the introduction to the method and results.

Weakness:
** The paper introduces new concepts like the Likert likelihood without a detailed theoretical discussion on its limitations or potential drawbacks, which could leave readers with questions on its applicability in various contexts.

**Questions For Authors:**

Handling of Noisy or Inconsistent Feedback:
How does your model handle situations where human feedback is noisy or inconsistent? Are there any mechanisms in place to detect or mitigate such issues, and how does this affect the model's overall performance in such cases?
Generalizability to Other Feedback Types:
Have you considered testing the proposed framework on other types of human feedback (e.g., response times, eye-tracking data, or even more complex preference signals)?

Again, I'm only familiar with a few works in this domain, if my question is reasonable, please let me know.

**Relation To Broader Scientific Literature:**

Gaussian Processes (GPs): The proposed method of using multiple likelihoods in a variational framework builds on previous works such as variational GPs (Hensman et al., 2015) and approximate inference schemes (Kuss & Rasmussen, 2005).

**Theoretical Claims:**

The ELBO formulation for mixed likelihoods appears mathematically sound based on standard variational GP methods.

---

> ### Author Rebuttal · Authors · 2025-03-31
>
> We thank the reviewer for their constructive comments. Below we provide our response.
>
> > The paper introduces new concepts like the Likert likelihood without a detailed theoretical discussion on its limitations or potential drawbacks, which could leave readers with questions on its applicability in various contexts.
>
> One limitation is that the Likert scale likelihood introduces additional hyperparameters (the cut points). Thus, this may increase the risk of overfitting in the case of limited data. A concrete example is subject #2 in Section 5.2, where the mixed likelihood-trained model fails to improve the performance. This is because subject #2 is predominantly confident and has a sharply different confidence distribution from other subjects; see Figure 9 in the appendix. In particular, they reported no ratings \\(\leq 3\\) at all. As a result, the Likert scale likelihood struggles to estimate the cut points.
>
> > Handling of Noisy or Inconsistent Feedback: How does your model handle situations where human feedback is noisy or inconsistent? Are there any mechanisms in place to detect or mitigate such issues, and how does this affect the model's overall performance in such cases?
>
> 1. The Bernoulli likelihood (and the preference likelihood) models the response as a Bernoulli distribution, which handles noise through the stochasticity in the likelihood by design.
> 1. The Likert scale likelihood Eq (2) also handles noise by stochasticity, as the discrete distribution Eq (2) assigns nonzero probabilities to all possible options.
> 1. In addition, the categorical distribution Eq (2) outputted by the Likert scale likelihood is damped with a uniform distribution using a lapse rate for enhanced robustness to outliers; see Line 666 in the appendix.
> 1. By their nature, all real-world data used in this paper are inherently noisy, as human responses are often inconsistent. The fact that our models improve the performance across different tasks already demonstrates their abilities to handle noise.
>
> > Generalizability to Other Feedback Types: Have you considered testing the proposed framework on other types of human feedback (e.g., response times, eye-tracking data, or even more complex preference signals)?
>
> We have implemented a response time likelihood and tested it on the robot gait data released by Shvartsman et al. (2024). The likelihood is based on the assumption that the response time \\(t\\) follows a log normal distribution. We observe that our response time likelihood does improve the performance (lower Brier scores and higher F1 scores). However, it was not fully developed before the submission deadline and thus we did not include it in the paper.
>
> https://imgur.com/a/qezbyeD
>
> Shvartsman, M., Letham, B., Bakshy, E., & Keeley, S. L. (2024). Response time improves gaussian process models for perception and preferences. In The 40th Conference on Uncertainty in Artificial Intelligence.

---

### Decision · Program_Chairs · 2025-05-01

**Decision:**

Reject

**Comment:**

Three reviewers recommend Weak Accept, and another reviewer recommends Accept. After reading the paper, the reviews and the rebuttal, I think the paper is very closely related to previous models in the GP literature that address some form of mixture of likelihood functions using GPs, such as "Heterogeneous Multi-output Gaussian Processes" or "Modular Gaussian Processes for Transfer Learning". In line with Reviewer f9ai, the paper applies straightforward stochastic variational inference to a specific combination of likelihoods. These are all well-known in the GP literature, and from that viewpoint, there is no really any methodological contribution. The applications of the proposed model, including active learning and human-in-the-loop, are one of the strengths of the paper, but without considering existing baselines, including baselines beyond GPs, the overall contribution is weakened.